# Cluster-Based Structural Redundancy Identification for Neural Network Compression

**DOI:** 10.3390/e25010009

**Published:** 2022-12-21

**Authors:** Tingting Wu, Chunhe Song, Peng Zeng, Changqing Xia

**Affiliations:** 1State Key Laboratory of Robotics, Shenyang Institute of Automation, Chinese Academy of Sciences, Shenyang 110016, China; 2Key Laboratory of Networked Control Systems, Chinese Academy of Sciences, Shenyang 110016, China; 3Institutes for Robotics and Intelligent Manufacturing, Chinese Academy of Sciences, Shenyang 110169, China; 4University of Chinese Academy of Sciences, Beijing 100049, China

**Keywords:** model compression, structure pruning, neural network acceleration, edge intelligence

## Abstract

The increasingly large structure of neural networks makes it difficult to deploy on edge devices with limited computing resources. Network pruning has become one of the most successful model compression methods in recent years. Existing works typically compress models based on importance, removing unimportant filters. This paper reconsiders model pruning from the perspective of structural redundancy, claiming that identifying functionally similar filters plays a more important role, and proposes a model pruning framework for clustering-based redundancy identification. First, we perform cluster analysis on the filters of each layer to generate similar sets with different functions. We then propose a criterion for identifying redundant filters within similar sets. Finally, we propose a pruning scheme that automatically determines the pruning rate of each layer. Extensive experiments on various benchmark network architectures and datasets demonstrate the effectiveness of our proposed framework.

## 1. Introduction

Compared with traditional machine learning methods, deep learning-based methods have greatly improved the performance of many computing tasks, such as image recognition [1], object detection [2] and speech segmentation [3]. Edge AI [4,5,6] stands out as a disruptive technology for 6G by embedding model training and inference capabilities at the edge of the network, which seamlessly integrates perception, communication, computing and intelligence to improve the efficiency, effectiveness, privacy and security of 6G networks. However, in pursuit of better performance, it is often at the expense of increasing computing power. It has gradually increased from the earliest LeNet [7] to approximately 20 layers of VGG [8]; in particular, the commonly used ResNet [9] and DenseNet [10] have increased astonishingly to hundreds of layers. The over-parameterization and redundant computation of the network makes it difficult to deploy on edge devices with limited computing resources. At present, model compression techniques for how to obtain a more efficient network have been proposed successively, including knowledge distillation [11], quantization [12,13], low-rank decomposition [14] and network pruning [15,16,17,18]. Among them, network pruning, which can dynamically evolve the baseline network into a more efficient sub-network, has become a widely recognized model compression method.

Network pruning is mainly divided into unstructured pruning and structured pruning. Unstructured pruning prunes individual weights in the model to compress the DNN (Deep Neural Network) [19]. However, although this method can greatly reduce the parameters, the generated unstructured matrices require a special sparse matrix operation library, which limits its practical acceleration in the general CNN acceleration framework. Structured pruning uses convolution kernels/channels or layers as pruning granularity for pruning. It has received widespread attention because of its advantages of being directly compatible with current general-purpose hardware and highly efficient basic linear algebra subprogram (BLAS) libraries. The research in this paper belongs to the category of structured pruning.

Typical filter pruning includes three stages [20]: (1) training a large, over-parameterized model (sometimes a pre-trained model), (2) pruning the trained large model according to certain criteria, and (3) fine-tuning the pruned model to restore the lost performance. Although existing pruning methods have achieved good results, there are still many problems. To assess the importance of filters, recently, a variety of filter pruning methods have been proposed to design more effective pruning guidelines, such as the average percentage of zero values (APoZ) [17], L1-norm [18], Taylor expansion [21], sparsity norm [22], geometric median (FPGM) [23], high rank (Hrank) [24] and variants of the pruning mask [25]. Due to the different distributions of the values of the convolution kernels in different layers, the abovementioned pruning methods based on global or local criteria for sorting filters may ignore filters with smaller values in the sorting but extract edge features. Huang et al. [26] compared different pruning standards and found that they have strong similarities, and that the importance of the obtained filters is almost the same, resulting in similar pruning structures. Recent work shows that the pruning structure is the key to determining the performance of the pruning model rather than the inheritance weight. Manually setting the pruning rate of each convolutional layer is equivalent to redesigning the network structure completely, and improper pruning rate settings will result in insufficient pruning or excessive pruning. In addition, for large networks, it is very expensive to accurately calculate the importance of the filters and set the pruning rate of each layer.

In this paper, we propose a clustering-based dynamic pruning method considering the similarity between filters. Compared with existing importance-based methods, we analyze the relevant information on the representational power among all filters within a layer and remove filters with overlapping functions. The proposed scheme takes into account edge features that are ignored based on importance ranking. Specifically, we cluster all filters within a layer in units of filters and select one deletion in each group whose features can be replaced by other filters. Each layer automatically generates groups according to the parameter distribution to determine the pruning rate of each layer, avoiding the problem of manually specifying the pruning rate. To assess the similarity of the representational power of filters in each group, a criterion is defined to measure the relative importance of all filters within a group. This criterion is scoped to a group and avoids the problem of threshold specification for global or local pruning. Extensive experiments demonstrate that our proposed method is more general than importance-based methods. Figure 1 shows a graphical illustration of our motivation and pruning framework. To summarize, our main contributions are as follows:We propose a novel pruning scheme that does not depend on importance but is based on the similarity between filters for channel-level pruning.We introduce an effective method for measuring the relative importance of filters, avoiding the problems of over-pruning and under-pruning caused by threshold specification.The proposed scheme automatically determines each layer’s pruning rate according to each layer’s parameter distribution, which avoids the problem of unreasonable pruning structure caused by manually specifying the pruning rate.A large number of experiments prove the effectiveness of the algorithm proposed in this paper.

## 2. Related Work

### 2.1. Unstructured Pruning

The research on network pruning originated from the 1989 paper on skeletonization [27]. LeCun [16] in 1990 and Hassibi [17] in 1993 proposed OBD and OBS methods, respectively, which measure the importance of the weight in the network based on the second derivative of the loss function relative to the weight (Hessian matrix). Hang Song et al. published a series of works on model compression for deep neural networks [28,29]. Among them, [29] compressed the classical networks AlexNet and VGG at that time, combined with various methods, such as pruning, quantization and Huffman coding to compress the network size by dozens of times. However, unstructured weight pruning does not guarantee GPU acceleration.

### 2.2. Structured Pruning

To overcome the above limitations, SSL [18] proposed to regularize the structures (i.e., filters, channels, filter shapes and layer depth) of DNNs. This was the first work to actually measure GPU acceleration, and structured pruning gradually became the focus of pruning research. Studies on structured pruning have been proposed one after another. The simplest of them is the magnitude-based weight pruning, which evaluates the importance according to the absolute value of the parameter or feature output [18,22,30,31]. Some studies [22,32,33] considered the impact of pruning on model loss as a criterion for measuring the importance of parameters. For example, Molchanov [21] was also based on Taylor expansion, using the first-order term’s absolute value in the objective function’s expansion relative to the activation function as pruning criteria. In addition, [34,35] considered the effect of pruning on the re-constructability of feature output, which minimizes the reconstruction error of the pruned network for feature output.

There are also other criteria based on the weights of the importance of ranking. He et al. [23] proposed a filter pruning via the geometric median (FPGM) method, the basic idea of which was to remove redundant parameters based on the geometric median. Lin et al. [24] developed a mathematically formulated method to prune filters with low-rank feature maps. The disadvantage of the above greedy algorithm is that it can only find the optimal local solution, which ignores the relationship between parameters. Some studies [36,37,38] try to consider the relationship between parameters, trying to find a better global solution. For example, Peng [38] proposed the collaborative channel pruning (CCP) method, which considers the dependencies between channels, formalizes the channel selection problem as a quadratic programming problem under constraints, and then uses sequential quadratic programming to solve it.

### 2.3. Other Compression Techniques

Other types of DNN model compression techniques are also being explored. Quantization [39,40] compressed the model by reducing the size of the weights or activations. XNOR-Net [41] and BinaryNet [42] used binary weights and activations to compress the model; [43,44] studied how to choose the appropriate quantization parameters to minimize the impact on the accuracy as much as possible; [45,46] explored how to make the distribution of quantified objects more suitable for quantification, and [47] introduced quantization operations during training to explore more efficient training of low-precision quantization networks. The quantization of ultra-low precision [48] and hybrid precision [49] has also been a popular topic in recent years. Knowledge distillation trains another simple network by using the output of a pre-trained complex network as a supervisory signal. The studies in [50,51] improved the prediction performance of the student model by adjusting the temperature; [52] transferred the knowledge of multiple teachers to a single student model so that the trained student model could handle the original tasks of the multiple teacher models, and [53] used different knowledge forms including output feature, intermediate feature, relational feature and structural feature. The low-rank decomposition sparse convolution kernel matrix was created by merging dimensions and imposing low-rank constraints. Since most of the weight vectors are distributed in the low-rank subspace, the convolution kernel matrix can be reconstructed with a few basis vectors to reduce the storage space. Jaderberg et al. [54] decomposed the convolution kernel of w×h into w×1 and 1×h, and reconstructed the learned dictionary weight linearly to obtain the output feature map; Liu et al. [8] used a two-stage decomposition method to study the redundancy between the channels; Wang et al. [55] proposed fixed-point decomposition, and then restored the performance through pseudo-full-precision weight repetition, weight balancing and fine-tuning; Kim et al. [56] proposed tucker decomposition, which performs binary decomposition of the first tensor along the input channel dimension to obtain convolutions of w×1, 1×h and 1×1, and Lebedev et al. [57] proposed CP decomposition on the basis of ternary decomposition.

The current model compression methods through pruning still determine the importance of a single parameter or filter by looking for a criterion and combining the pruning method and processing to restore the performance of the pruned model. Different from previous methods, we use clustering to find filters with overlapping functions more effectively by comparing the similarity between filters.

## 3. Methodology

In this section, we propose a novel pruning scheme, which is the cluster similarity-based filter pruning method. We first introduce the overall framework and related notation definitions, and then describe the motivation and implementation details. Finally, we propose the corresponding pruning scheme.

### 3.1. Overall Framework

We show the flow of a single pruning of the proposed pruning scheme in Figure 1. First, we use a filter as the unit to cluster all the convolution kernels in one layer and obtain the cluster label corresponding to each filter. We improved the single-threshold selection from the previous single-dimensional space to map the filter into the multi-dimensional space through clustering. The clustering results are used to guide the determination of redundant filters in the next step. Since filters with similar features are in the same cluster set, we only need to find redundant filters in each cluster space. Then, redundant filters in each cluster are obtained according to the proposed criterion and removed. Finally, after several iterations, we can get the final compact network structure without specifying the pruning rate of each layer and reach the specified pruning rate constraint.

### 3.2. Motivation and Definitions

One of the main problems of filter pruning is how to select effective filters and retain as much of the expressive power of the original network as possible. Current amplitude-based pruning algorithms all rely on the assumption that removing relatively insignificant weights in the network has little effect on the pruned network performance. Unlike current views based on parameter importance, we propose that the removal of any one of the filters will not significantly impair the representational power of the network as long as there are two sufficiently similar channels. This reduction also resonates with the well-known Hebbian principle, which roughly states that “neurons which fire together, wire together”. The visualization results of the filters and feature maps of the first convolutional layer of VGG16 are shown in Figure 2. It can be seen that there are a large number of similar filters in the trained network.

Assuming that the neural network has L convolutional layers, $N_{l}$ and $N_{l+1}$ represent the number of input channels and output channels of the $l_{th}$ layer convolution layer, respectively. $F^{\left(l,i\right)}$ represents the $i_{th}$ filter of the $l_{th}$ layer, and the corresponding input feature map can be expressed as $F^{\left(l,i\right)}\in R^{H\times W\times B}$, where H,W,B represent the height and width of the feature maps and the batch size, respectively. The tensor of the connections of the $l_{th}$ and $l+1_{th}$ layers can be parameterized by $W\in R^{N_{l}\times N_{l+1}\times K\times K},1\leq l\leq L$.

We demonstrate that if there is a set of similar filters within a layer, pruning filters randomly or selectively in that set is better than pruning the least important filters within that layer. We assume that there is a set of similar filters $S_{\alpha }$ and another $S_{\beta }$ with remaining filters in the $l_{th}$ layer, containing n and m filters, respectively. We choose positive constants a,b>0 and use the random events $\left(\sum_{i=1}^{n}\alpha_{i}\geq a\right)$ and $\left(\sum_{i=1}^{m}\beta_{i}\geq b\right)$ to indicate that the filters in $S_{\alpha }$ and $S_{\beta }$ perform better, and use the sum of the two to indicate the performance of the entire layer. When removing a filter from the $l_{th}$ layer, there are several situations including (1) no pruning; (2) randomly selecting a filter to prune in $S_{\alpha }$; (3) pruning according to the minimum rule in $S_{\alpha }$; (4) pruning according to the minimum rule in $S_{\beta }$; (5) pruning the least important filter in the layer, i.e., $\min(S_{\alpha },S_{\beta })$:(1)$$p_{o}=P(\sum_{i=1}^{n}\alpha_{i}\geq a)+P(\sum_{i=1}^{m}\beta_{i}\geq b)$$
(2)$$p_{\alpha r}=P(\sum_{i=1}^{n-1}\alpha_{i}\geq a)+P(\sum_{i=1}^{m}\beta_{i}\geq b)$$
(3)$$p_{\underline{\alpha }}=P(\sum_{i=1}^{n}\alpha_{i}-\underline{\alpha }\geq a)+P(\sum_{i=1}^{m}\beta_{i}\geq b)$$
(4)$$p_{\underline{\beta }}=P(\sum_{i=1}^{n}\alpha_{i}\geq a)+P(\sum_{i=1}^{m}\beta_{i}-\underline{\beta }\geq b)$$
(5)$$p_{g}=\frac{n}{m+n}p_{\underline{\alpha }}+\frac{m}{m+n}p_{\underline{\beta }}$$

Note that $0\leq \alpha_{n}-\underline{\alpha }\leq \alpha_{n}$; therefore, we have
(6)$$P(\sum_{i=1}^{n-1}\alpha_{i}\geq a)\leq P(\sum_{i=1}^{n}\alpha_{i}-\underline{\alpha }\geq a)\leq P(\sum_{i=1}^{n}\alpha_{i}\geq a)$$
indicating that $p_{\alpha r}\leq p_{\underline{\alpha }}\leq p_{o}$. For any filter, the contribution to the network cannot be infinite, where the variance is uniformly bounded.
$$\exists C_{1}>0,s.t.D\eta_{i}\leq C_{1},i=1,2,\cdots ,n$$

By Chebyshev’s inequality, for any real number ϵ>0,
(7)$$P(\frac{1}{n}\left|\sum_{i=1}^{n}\left(\alpha_{i}-E\alpha_{i}\right)\right|\geq \epsilon )\leq \frac{D(\sum_{i=1}^{n}\alpha_{i})}{\epsilon^{2}n^{2}}$$

From Equation (7) we can get:$$\mathrm{Cov}\left(\alpha_{i},\alpha_{j}\right)\leq \sqrt{D\alpha_{i}\cdot D\alpha_{j}}\leq C_{1}$$

We assume that there are $C_{2}n\;(0\leq C_{2}\leq 1)$ pairs of similar filters in the set $S_{\alpha }$, i.e., $\#\{(i,j):\left.\mathrm{Cov}\left(\alpha_{i},\alpha_{j}\right)\neq 0,i\neq j,i,j=1,\cdots ,n.\right\}\leq C_{2}n$, then we have:(8)$$D(\sum_{i=1}^{n}\alpha_{i})=\sum_{i=1}^{n}D\alpha_{i}+\sum_{i\neq j}\mathrm{Cov}\left(\alpha_{i},\alpha_{j}\right)\leq C_{1}n+C_{1}C_{2}n=C_{1}\left(1+C_{2}\right)n$$

Available by Equation (8):(9)$$P(\frac{1}{n}\left|\sum_{i=1}^{n}\left(\alpha_{i}-E\alpha_{i}\right)\right|\geq \epsilon )\leq \frac{C_{1}\left(1+C_{2}\right)}{\epsilon^{2}n}\to 0$$

This means $\frac{1}{n}\sum_{i=1}^{n}\left(\alpha_{i}-E\alpha_{i}\right)$ converges in probability to zero, i.e., $\frac{1}{n}\sum_{i=1}^{n}\left(\alpha_{i}-E\alpha_{i}\right)\overset{P}{\to }0$. We consider the filter’s contribution to be a positive number that expects a uniform positive lower bound:$$\exists \epsilon_{0}>0,\;s.t.E\alpha_{i}\geq \epsilon_{0},i=1,2,\cdots ,n$$

By Equation (9) we can get:(10)$$\begin{array}{l} P(\frac{1}{n}\sum_{i=1}^{n}\left(\alpha_{i}-E\alpha_{i}\right)>-\frac{\epsilon_{0}}{2})=P(\sum_{i=1}^{n}\alpha_{i}>\sum_{i=1}^{n}E\alpha_{i}-\frac{\epsilon_{0}}{2}n)\\ =P(\sum_{i=1}^{n}\alpha_{i}>\frac{\epsilon_{0}}{2}n+\sum_{i=1}^{n}\left(E\alpha_{i}-\epsilon_{0}\right))\\ \leq P(\sum_{i=1}^{n}\alpha_{i}>\frac{\epsilon_{0}}{2}n)\leq P(\sum_{i=1}^{n}\alpha_{i}>b) \end{array}$$

Letting n→+∞, then $\frac{1}{n}\sum_{i=1}^{n}\left(\alpha_{i}-E\alpha_{i}\right)\overset{P}{\to }0$, and we have:(11)$$\underset{n\to \infty }{\lim}P(\sum_{i=1}^{n}\alpha_{i}>b)\geq \underset{n\to \infty }{\lim}P(\frac{1}{n}\sum_{i=1}^{n}\left(\alpha_{i}-E\alpha_{i}\right)>-\frac{\epsilon_{0}}{2})=1$$
(12)$$\;\underset{n\to \infty }{\lim}P(\sum_{i=1}^{n}\alpha_{i}-\alpha_{r}>b)=\underset{n\to \infty }{\lim}P(\sum_{i=1}^{n}\alpha_{i}-\underline{\alpha }>b)=1$$

Therefore, we have $p_{\alpha r}\approx p_{\underline{\alpha }}\approx p_{o}$ for an n that is large enough. Furthermore, $p_{\underline{\beta }}\leq p_{o}\approx p_{\underline{\alpha }}$, and since $p_{g}$ is the average of $p_{\underline{\alpha }}$ and $p_{\underline{\beta }}$, there is $p_{\underline{\beta }}\leq p_{g}\leq p_{\underline{\alpha }}$. In summary, we have $p_{\underline{\beta }}\leq p_{g}\leq p_{\alpha r}\leq p_{\underline{\alpha }}\leq p_{o}$, which indicates that pruning filters in a similar set (even randomly) works better than pruning the least important filter in that layer. This provides the basis for our pruning below. The next challenge is how to find similar sets and how to determine which filters within a set are redundant.

### 3.3. Cluster Pruning

K-means clustering provides a solution for how to find similar convolution kernels within a layer. For the $l_{th}$ layer, $N_{l+1}$ filters are divided into k clusters, then the clustering result is $\{S_{1},S_{2},S_{3},\cdots ,S_{k}\},k\leq N_{l+1}$, where $S_{1}=\{F_{1},F_{3},F_{7},F_{9},F_{12}\}$, $S_{2}=\{F_{2},F_{5},F_{14}\}$,…, $S_{k}=\{F_{6},F_{10},F_{13},F_{N_{l+1}}\}$. We divide each layer into different sets in which convolutional kernels contain similar feature information. At the same time, we also get the centroid $\left\{C_{1},C_{2},C_{3},\cdots ,C_{k}\right\}$, representing the set information:(13)$$C_{k}=\frac{\sum_{F_{i}\in S_{k}}F_{i}}{\left|S_{k}\right|}$$
where $C_{k}$ represents the center of the $k_{th}$ cluster, $\left|S_{k}\right|$ represents the number of objects in the $k_{th}$ cluster and $F_{i}$ represents the $i_{th}$ object in the $k_{th}$ cluster.

After getting sets containing similar information, the task is to find redundant filters in each set. The cluster center anchors the representative information of all filters in a similar cluster. After obtaining the cluster center of each similar cluster, it is intuitive to choose to keep the cluster center and delete other filters of the same cluster. However, this simple pruning method causes a large number of filters to be discarded, resulting in a sharp drop in performance. We look for a reasonable criterion for progressive pruning to determine redundant kernels in a cluster. In contrast to the intuitive pruning approach, we propose that cluster centers can be pruned. Since the cluster center is the mean value of all filters in clusters in each dimension, the cluster center of the remaining filters is still in the original cluster center position after removing the cluster center. This indicates that the information contained in all filters in the cluster is not lost after removing the cluster center.

The above claim is in the ideal state, that is, the centroid of each cluster is exactly one element in the set. In experiments, it is difficult to find the element in the cluster set that happens to be the cluster center for each clustering. We find a compromise scheme, where each time we find the element closest to the cluster center, the information represented by this element can be transferred to other filters, and the original cluster set still retains the original information. First, we calculate the distance between all filters and cluster center in the $k_{th}$ cluster set of the $l_{th}$ layer:(14)$$\begin{array}{l} D_{i}^{k}=dist(F_{i}^{k},C_{k}),1\leq i\leq \left|S_{k}\right|\\ =\sqrt{\sum_{n=1}^{N_{l+1}}\sum_{k_{1}=1}^{K}\sum_{k_{2}=1}^{K}{\left|W^{i}\left(n,k_{1},k_{2}\right)-C_{k}(n,k_{1},k_{2})\right|}^{2}} \end{array}$$
where $W^{i}\left(n,k_{1},k_{2}\right)$ is each weight parameter in the $i_{th}$ filter, $C_{k}(n,k_{1},k_{2})$ is each weight parameter in the cluster center of the $k_{th}$ cluster and $F_{i}^{k}$ is the $i_{th}$ filter in the $k_{th}$ cluster. Then, the filter that needs to be pruned in the similar cluster is:(15)$$P_{k}=\underset{1\leq i\leq \left|S_{k}\right|}{\arg\;\min\;}D_{i}^{k}$$
where $P_{k}$ is the filter to be pruned in the $k_{th}$ cluster. Finally, the set of all pruned filters in the $l_{th}$ layer is $\left\{P_{1}^{l},P_{2}^{l},P_{3}^{l},\cdots ,P_{k}^{l}\right\}$.

### 3.4. Pruning Scheme

In the above section, we propose how to identify redundant filters from a clustering set. However, for the number of hundreds or even thousands of filters in the current network structure, the number of redundant filters obtained by one clustering is far from enough. Therefore, we propose an iterative pruning scheme to identify more redundant filters to meet the compression requirements of large pruning rates and large network structures. After one pruning is completed, the next pruning continues to cluster to find a new similarity set and delete the redundant filters in the set. Compared with the current method of manually specifying the pruning rate of each layer, we only need to adjust one parameter to control the pruning rate of all layers. This is very efficient for large networks with hundreds of layers.

The overall workflow of our cluster-based pruning algorithm is shown in Algorithm 1 and can be summarized as follows:
For each convolutional layer, first initialize each cluster center $\left\{\mu_{1},\mu_{2},\mu_{3},\ldots ,\mu_{k}\right\}$ and compare any filter in the layer with each cluster center to construct a $N_{l+1}\times k$ distance matrix. In each iteration, $\lambda_{i}=\arg\min_{k\in \{1,2,3,\ldots ,k\}}d_{i,k}$ is obtained in each row, and the corresponding filter is divided into the corresponding cluster k, and, finally, the cluster set $S^{l}=\{S_{1}^{l},S_{2}^{l},S_{3}^{l},\ldots ,S_{k}^{l}\}$ and cluster center $C^{l}=\{C_{1}^{l},C_{2}^{l},C_{3}^{l},\ldots ,C_{k}^{l}\}$ are obtained.For each cluster set $S_{k}^{l}$ obtained, each filter in the set and the cluster center $C_{k}^{l}$ obtain a $\left|S_{k}^{l}\right|$-dimensional vector according to Equation (2), and the filter corresponding to the minimum value in the vector is determined as the layer that needs to be pruned filter.After one pruning, calculate the pruning end condition, that is, the amount of computation $rate_{FLOPs}$ or parameters $rate_{params}$ after pruning, prune in a loop until the given pruning rate is reached and fine-tune the generated model to restore performance.

**Algorithm 1**: Iterative pruning algorithm.**Input:** Training dataset D; the model with W, and each layer with $W\in R^{N_{l}\times N_{l+1}\times K\times K},1\leq l\leq L$; FLOPs or params pruning rate: $rate_{FLOPs}$**/**$rate_{params}$.**Output:** The pruned model $W^{P}$1: 
W←train(W,D)2: 
**while** pruned rate = 0 to rate **do**
3:      **for**
 l=1,2,…,L **do**4:           
initialize the clusters $\left\{\mu_{1},\mu_{2},\mu_{3},\ldots ,\mu_{k}\right\}$
5:           **for**
$i=1,2,\ldots ,N_{l+1}$ **do**
6:                $d_{i,k}={\left\|F_{i}-\mu_{k}\right\|}_{2}$7:                
$\lambda_{i}=\arg\min_{k\in \{1,2,3,\ldots ,k\}}d_{i,k}$8:                
$S_{\lambda_{i}}=S_{\lambda_{i}}\cup \{F_{i}\}$
9:           
**end for**
10:           
$S^{l}=\{S_{1}^{l},S_{2}^{l},S_{3}^{l},\ldots ,S_{k}^{l}\}$
11:           
$C^{l}=\{C_{1}^{l},C_{2}^{l},C_{3}^{l},\ldots ,C_{k}^{l}\}$

12:           
**for** k=1,2,…,k **do**
13:                
**for**$j=1,2,\ldots ,\left|S_{k}^{l}\right|$ **do** 

14:                     
$D_{j}^{k}=dist(F_{j}^{k},C_{k}^{l})$
15:                     
$P_{k}=\underset{1\leq i\leq \left|S_{k}^{l}\right|}{\arg\min}D_{j}^{k}$
16:                
**end for**
17:           
**end for**18:         
$P^{l}=\{P_{1}^{l},P_{2}^{l},P_{3}^{l},\cdots ,P_{k}^{l}\}$19:        
**end for** 20:       
$W^{P}\leftarrow W-P$
21: 
**end while**
22: 
$W\leftarrow finetune(W^{P},D)$

In the network structure, the processing of special structures, such as dense block, residual block and inverted residual block, also has a greater impact on the final compression performance. In these structures, we process in blocks and perform pruning on the basis of satisfying the original relative relationship. For example, in a residual block, the number of input channels and output channels of each block is the same, and there are also constraints on the number of three convolutional layers in the block. According to previous work [9], the 3 × 3 convolution kernel in the residual block has the same number of input and output channels, that is, the output channels of the previous layer and the input channels of the last layer are the same, as shown in Figure 3.

## 4. Experiments

### 4.1. Experimental Settings

We evaluate the effectiveness of our algorithm on CIFAR-10, CIFAR-100 [58] and ILSVRC-2012 [1] datasets using representative CNN architectures VGGNet [8] and ResNet [9]. CIFAR10 contains 50,000 training images and 10,000 testing images (size 32 × 32), which are categorized into 10 different classes. CIFAR100 is similar to CIFAR-10 but has 100 classes. ImageNet contains 1.28 million training images and 50 k validation images of 1000 classes. VGGNet and ResNet represent two typical network structures with single branch and multiple branches, respectively. All experiments are implemented on four NVIDIA TITAN Xp GPUs using PyTorch.

We measure the complexity of the network using floating point operations (FLOPs) required for forward propagation. The computational cost of one convolutional layer is:$$FLOPs=HW\left(C_{in\;}K^{2}+1\right)C_{out}$$
(16)$$Params=\left(C_{in\;}K^{2}+1\right)C_{out}$$
where H and W are the height and width of the input feature map of the layer, respectively, and $C_{in}$ and $C_{out\;}$ are the number of input channels and output channels. In this paper, we use the drop rate of FLOPs to evaluate the compression performance of each algorithm, that is, the smaller the accuracy drop of the compressed network model under the same compression ratio, the better the algorithm performance:$$rate_{FLOPs}=1-\frac{FLOPs_{original}}{FLOPs_{compressed\;}}$$
(17)$$rate_{params}=1-\frac{Params_{original}}{Params_{compressed\;}}$$

### 4.2. Results on CIFAR-10/100 Datasets

We evaluate the effectiveness of the proposed framework using VGG16 [8] and ResNet-32/56/110 [9] on CIFAR10 and CIFAR100 datasets [58] and compare with existing algorithms, such as L1 [18], Molchanov et al. [21], SFP [59], FPGM [23], Hrank [24] and SRR-GR [60]. All the networks are trained using SGD with Nesterov momentum [61] of 0.9, a weight decay parameter of $10^{-4}$ and an initial learning rate of 0.1. The learning rate is set to 0.001 when updating parameters or fine-tuning. For VGG16, the baseline network is trained for 300 epochs with a batch size of 256. For ResNet, the baseline network is trained for 200 epochs with a batch size of 256.

It can be seen from Table 1 that our proposed pruning framework achieves less accuracy loss with higher computational compression using VGG16 on the CIFAR10 and CIFAR100 datasets. We find effective channels in the network and reduce the false deletion of channels to achieve better performance than other algorithms. Comparing the results on the CIFAR10 and CIFAR100 datasets, the same network has different redundancy on different datasets, as shown in Figure 4. Due to the large redundancy of VGG16 on the CIFAR10 dataset, our performance differs little from other algorithms when the compression ratio is small. However, when the compression ratio becomes larger, the performance of each algorithm is significantly different. For example, when the compression ratio reaches 90% on CIFAR10, the performance of our algorithm and Hrank is quite different, that is, we identify redundant channels more effectively. However, since VGG16 has less redundancy on CIFAR100, pruning is more difficult. When the pruning rate is approximately 50%, the network performance loss after pruning is obvious; however, we still maintain good performance at larger compression ratios.

The results of ResNet with different depths on the CIFAR10 and CIFAR100 datasets are shown in Table 2. Pruning is more challenging due to the residual structure in ResNet. In addition to the influence of redundant judgment criteria, the processing of residual structures also affects the pruning performance of ResNet. As can be seen from the table, we reduce the accuracy loss while reaching the same or higher compression ratio in ResNet of different depths. Due to the overfitting of the network on the CIFAR10 dataset, the accuracy of pruned ResNets of different depths does not decrease but increases after compression. For example, our framework compresses 60.1% of the FLOPs on the ResNet-56, but the accuracy increased by 0.54%. The uncompressed ResNet performs generally on the CIFAR100 dataset, our algorithm still maintains the original performance after compressing half of the parameters. However, interestingly, the redundancy of ResNet on the CIFAR100 dataset does not increase with network depth, for example, ResNet-110 still has a larger accuracy loss than ResNet-56 with less pruning rate.

### 4.3. Results on ILSVRC-2012

In the experiments, we use ResNet-18/34/50 to demonstrate the proposed pruning performance on a large-scale dataset, ILSVRC-2012. All the baseline networks are obtained by training 100 epochs with a batch size of 256. We follow the same parameter settings as [20,60]. We compare the proposed method with ThiNet [34], FPGM [23], MIL [62], L1 [18], CP [35], SFP [59], Hrank [24], PGMPF [25] and SRR-GR [60]. All the results of the other methods in the table are directly from their reports in the literature.

The results of ResNet with different depths on ILSVRC-2012 are shown in Table 3. It can be seen from the table that our framework still has good compression performance on large datasets. We reduce the accuracy loss at the same computational compression ratio. Combined with the analysis of the above CIFAR100 dataset, it shows that the network is more sensitive to compression on larger datasets that are underfitting. When the compression ratio increases, the accuracy drops significantly, and the performance of each algorithm varies significantly. For example, on ResNet-18, when other algorithms compress less than half of the calculation, the Top-1 accuracy drops by between 2 and 4%, but ours controls the accuracy loss to within 2%. However, it is common sense that the deeper the network, the greater the redundancy of the model. Under the same compression rate, the accuracy of ResNet-50 only drops by 0.35%, and ResNet-18 drops by 1.82%.

## 5. Conclusions

Aiming at the problems of ignoring edge features and manually specifying the pruning rate in current importance-based model pruning algorithms, this paper proposes a new model pruning framework based on similarity clustering. We reconsider the redundancy of neural network models from the perspective of similarity and find similar sets of filters through clustering and then propose a criterion for determining filter redundancy in similar sets. In order to solve the problem of a long compression period caused by excessive fine-tuning, we propose a corresponding iterative pruning scheme. Extensive experiments demonstrate the effectiveness of our proposed compression framework, while we cluster filters into multidimensional spaces and reconsider filter redundancy from a similar perspective, without specifying pruning rate. However, multi-dimensional values for clustering are still an unsolved problem. In the next step, we will combine this research with reinforcement learning and continue to mine redundant parameters in the network.

## Figures and Tables

**Figure 1 entropy-25-00009-f001:**
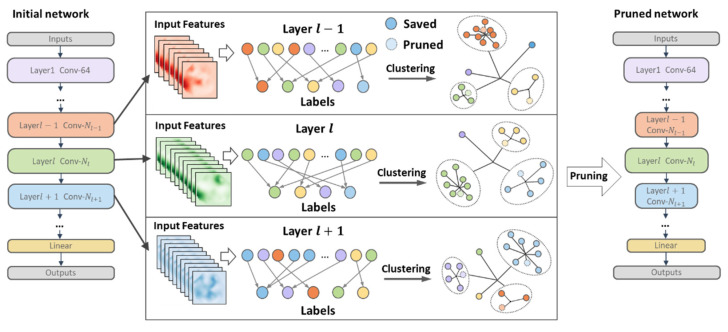
Our proposed pruning framework. First, we traverse layer-by-layer, and use the convolution layer as the unit to cluster the filters in the convolution layer, respectively. After clustering, each filter gets its label and grouping, and redundant filters are removed in the similarity group according to the proposed redundancy criterion.

**Figure 2 entropy-25-00009-f002:**
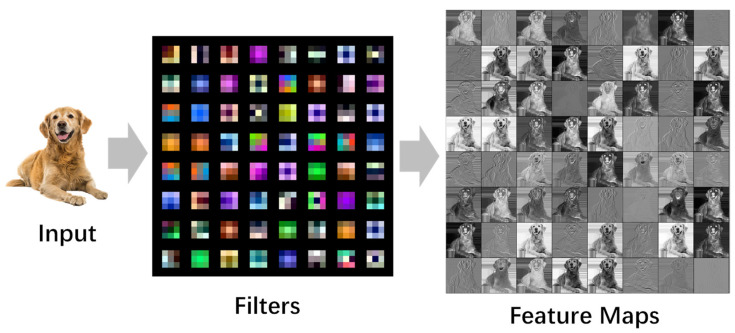
Visualization of filters and feature maps of the first convolutional layer of VGG16. As can be seen from the figure, a convolutional layer has multiple filters with similar expressive abilities, and the feature maps obtained by similar filters after convolution are also similar.

**Figure 3 entropy-25-00009-f003:**
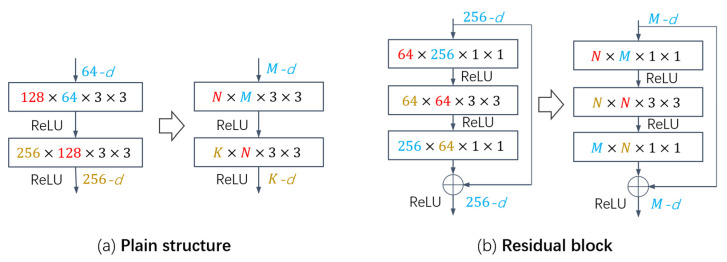
Pruning of different structures: (**a**) is a plain structure, where the number of pruned channels between the two layers is not constrained; (**b**) is a residual block, where each block has the same number of input and output channels, and the number of input and output channels of the middle layer and the upper and lower layers both satisfy specific constraints.

**Figure 4 entropy-25-00009-f004:**
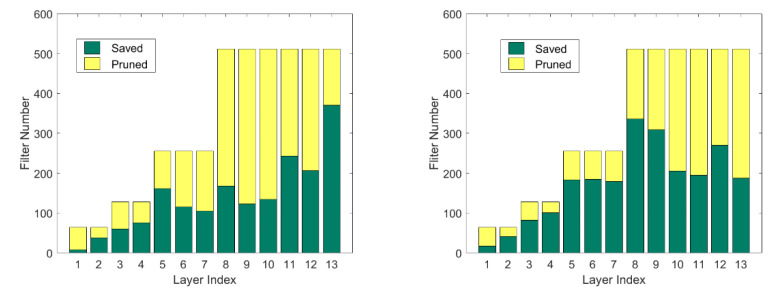
Our method uses VGG16 on the CIFAR10/100 dataset to obtain different pruned structures according to different redundancy.

**Table 1 entropy-25-00009-t001:** Comparison of Pruned VGG16 on CIFAR10/100 Datasets.

Model/Data	Method	Baseline Top-1 Acc (%)	Pruned Top-1 Acc (%)	Top-1(↓) Acc (%)	FLOPs (↓) (%)	Params (↓) (%)
VGG16/CIFAR10	L1	93.58	93.31	0.27	34.20	64.00
	FPGM	93.58	93.23	0.34	34.20	64.00
	**Ours**	**93.92**	**93.70**	**0.22**	**40.98**	**42.46**
	Taylor	93.92	91.24	2.78	78.03	84.56
	Hrank	93.96	91.23	2.73	76.50	92.00
	**Ours**	**93.92**	**92.49**	**1.43**	**87.49**	**91.20**
VGG16/CIFAR100	L1	73.45	71.21	2.24	50.44	50.23
	Taylor	73.45	70.34	2.36	51.48	59.89
	FPGM	73.45	71.39	2.06	-	48.93
	**Ours**	**73.45**	**71.91**	**1.54**	**54.11**	**62.49**

**Table 2 entropy-25-00009-t002:** Comparison of Pruned ResNet on CIFAR10/CIFAR100 Datasets.

Model/Data	Method	Baseline Top-1 Acc (%)	Pruned Top-1 Acc (%)	Top-1 (↓) Acc (%)	FLOPs (↓) (%)
ResNet-32/CIFAR10	L1	91.82	80.01	11.81	43.76
	SFP	91.33	91.60	+0.27	53.16
	FPGM	91.33	91.90	+0.57	53.16
	**Ours**	**91.82**	**92.11**	**+0.29**	**55.36**
ResNet-56/CIFAR10	L1	93.04	91.31	1.75	27.60
	SFP	93.59	92.26	1.33	52.60
	FPGM	93.59	92.89	0.70	52.60
	HRank	93.26	93.17	0.09	50.00
	SRR-GR	93.38	93.75	+0.37	53.80
	**Ours**	**92.55**	**93.09**	**+** **0.54**	**60.10**
ResNet-110/CIFAR10	L1	93.53	92.94	0.61	38.60
	SFP	93.68	93.38	0.30	40.80
	FPGM	93.68	93.73	+0.05	52.30
	Hrank	93.50	92.65	0.85	68.60
	**Ours**	**93.60**	**93.17**	**0.43**	**70.59**
ResNet-32/CIFAR100	L1	66.48	58.11	8.37	43.76
	SFP	66.48	64.27	2.21	53.16
	FPGM	66.48	66.64	0.16	53.16
	**Ours**	**66.48**	**66.87**	**+0.39**	**50.51**
ResNet-56/CIFAR100	SFP	69.08	68.03	1.05	63.16
	FPGM	69.08	67.75	1.33	63.16
	PGMPF	72.92	70.21	2.71	52.6
	**Ours**	**69.08**	**68.57**	**0.51**	**63.48**
ResNet-110/CIFAR100	**Ours**	**71.26**	**70.28**	**0.98**	**57.73**

**Table 3 entropy-25-00009-t003:** Comparison of Pruned ResNet on ImageNet.

Model/Data	Method	Baseline Top-1 Acc (%)	Pruned Top-1 Acc (%)	Top-1 (↓) Acc (%)	Baseline Top-5 Acc (%)	Pruned Top-5 Acc (%)	Top-5 (↓) Acc (%)	FLOPs (↓) (%)
ResNet-18	MIL	69.98	66.33	3.65	86.94	89.24	2.30	34.6
	SFP	70.28	67.10	3.18	89.63	87.78	1.85	41.8
	FPGM	70.28	67.81	2.47	89.63	88.11	1.52	41.8
	PGMPF	70.23	66.67	3.56	89.51	87.36	2.15	53.5
	**Ours**	**70.48**	**68.66**	**1.82**	**89.60**	**88.44**	**1.16**	**53.8**
ResNet-34	MIL	73.42	72.99	0.43	91.36	91.19	0.17	24.8
	L1	73.23	72.17	1.06	-	-	-	24.2
	SFP	73.92	71.83	2.09	91.62	90.33	1.29	41.1
	FPGM	73.92	72.11	1.81	91.62	90.69	0.93	41.1
	PGMPF	73.27	70.64	2.63	91.43	89.87	1.56	52.7
	**Ours**	**73.90**	**72.55**	**1.35**	**91.59**	**90.79**	**0.80**	**52.1**
ResNet-50	ThiNet	75.30	74.03	1.27	92.20	92.11	0.09	36.79
	SFP	76.15	74.61	1.54	92.87	92.06	0.81	41.8
	FPGM	76.15	75.03	1.12	92.87	92.40	0.47	42.2
	HRank	76.15	74.98	1.17	92.87	92.33	0.54	43.76
	SRR-GR	76.13	75.76	0.37	92.86	92.60	0.19	44.10
	PGMPF	76.01	75.11	0.90	92.93	92.41	0.52	53.5
	**Ours**	**75.82**	**72.47**	**0.35**	**92.95**	**92.68**	**0.27**	**53.1**

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
