# Peer review of "Cluster-Based Structural Redundancy Identification for Neural Network Compression"

_entropy, 2022, doi:10.3390/e25010009_

Round 1

Reviewer 1 Report

Accepted on the present form, good research article.

Author Response

Dear Reviewer,

Thank you very much for supporting my work!

Best wishes for you

Reviewer 2 Report

Please revise your manuscript according to my comments listed in the attached file.

Author Response

Dear Reviewer,

Thank you for your letter and the comments concerning our manuscript entitled "Cluster-based Structural Redundancy Identification for Neural Network Compression". We have studied the comments carefully and revised our manuscript based on these comments. The main changes have been highlighted in the revised manuscript and listed in this response letter for your convenience.

Comment 1: Page 1, Abstract, Lines 13-14: The following sentence demonstrates the case when the meaning is not conveyed due to the lack of one punctuation character comma. Please put a comma at the end of the first message “Existing works typically compress models based on importance”, separating it from the second one with additional clarification: “removing unimportant filters”. The correct sentence should look like this: “Existing works typically compress models based on importance, removing unimportant filters.”

Response: Thanks for the valuable comment. The mistake has been fixed and highlighted in the revised paper. The relevant content in the paper is as follows:

        “Existing works typically compress models based on importance, removing unimportant filters.”

Comment 2: Page 1, Line 43: Please define the abbreviation DNN at its first use and, if possible, the DNN literature reference in the following Unstructured pruning prunes individual weights in the model to compress the DNN.

Response: Thanks for the valuable comment. In the revised manuscript, We have added the definition of DNN and cited related literature.

        “Unstructured pruning prunes individual weights in the model to compress the DNN (Deep Neural Network) [19].”

[19]  G. E. Hinton, S. Osindero, and Y.-W. Teh, “A fast learning algorithm for deep belief nets,” Neural computation, vol. 18, no. 7, pp. 1527–1554, 2006.

Comment 3: There are plenty of incorrect reference numbers found in the manuscript text. Please correct all of them. There is a shift in part of the reference numbers, so please be careful in setting the reference numbers in your revised paper. I can show you several examples to be corrected in your revision. Please check all other reference numbers in the manuscript text.

Response: Thanks for the valuable comment. We apologize for the mistake. Since we added a reference in the second comment, we adjusted the numbering of all references. In addition, we have checked all reference numbers and corrected the wrong reference numbers in the revised manuscript.

Reviewer 3 Report

The manuscript presents a novel model compression method from the perspective of structural redundancy, which greatly reduces the computational and parameter quantities of the model while ensuring the performance of the original network. The manuscript is well written and easy to follow. However, there are still many concerns that should be addressed as follows:

1. In Algorithm 1, rate_FLOPs/rate_params is used as constraint, but the calculation method of rate_params is not mentioned in the paper.

2. In the experimental setting, how to choose the initial number of clusters of K-means?

3. What is the number of epochs that need to be fine-tuned after each pruning in the experiment?

4. Some cites to pictures and tables need to correspond to each other, such as #169: “…the proposed pruning scheme in Figure 2” should be “Figure 1”.

5. The pruned and saved channels in Figure 1 need to be marked to make the figure easier to read.

Author Response

Dear Reviewer,

Thank you for your letter and the comments concerning our manuscript entitled "Cluster-based Structural Redundancy Identification for Neural Network Compression". We have studied the comments carefully and revised our manuscript based on these comments. The main changes have been highlighted in the revised manuscript and listed in this response letter for your convenience.
